# Patient Safety Attitudes among Doctors and Nurses: Associations with Workload, Adverse Events, Experience

**DOI:** 10.3390/healthcare10040631

**Published:** 2022-03-27

**Authors:** Khaild AL-Mugheed, Nurhan Bayraktar, Mohammad Al-Bsheish, Adi AlSyouf, Mu’taman Jarrar, Waleed AlBaker, Badr K. Aldhmadi

**Affiliations:** 1Surgical Nursing Department, Faculty of Nursing, Near East University, Nicosia 99138, Cyprus; nurhan.bayraktar@neu.edu.tr; 2Healthcare Administration Department, Batterjee Medical College, Jeddah 21442, Saudi Arabia; mohammed.ghandour@bmc.edu.sa; 3Department of Managing Health Services and Hospitals, Faculty of Business Rabigh, College of Business (COB), King Abdulaziz University, Jeddah 21991, Saudi Arabia; oal@kau.edu.sa; 4Vice Deanship for Quality and Development, College of Medicine, Imam Abdulrahman Bin Faisal University, Dammam 34212, Saudi Arabia; mkjarrar@iau.edu.sa; 5Medical Education Department, King Fahd Hospital of the University, Al-Khobar 34445, Saudi Arabia; 6Department of Internal Medicine, College of Medicine, Imam Abdulrahman Bin Faisal University, Dammam 34212, Saudi Arabia; wialbakr@iau.edu.sa; 7Department of Health Management, College of Public Health and Health Informatics, University of Ha’il, Ha’il 81451, Saudi Arabia; b.aldhmadi@uoh.edu.sa

**Keywords:** Safety Attitudes Questionnaire, workload, adverse events, experience, patient safety

## Abstract

Patient safety concept has achieved more attention from healthcare organizations to improve the safety culture. This study aimed to investigate patient safety attitudes among doctors and nurses and explore associations between workload, adverse events, and experience with patient safety attitudes. The study used a descriptive cross-sectional design and the Turkish version of the Safety Attitudes Questionnaire. Participants included 73 doctors and 246 nurses working in two private hospitals in Northern Cyprus. The participants had negative perceptions in all patient safety domains. The work conditions domain received the highest positive perception rate, and the safety climate domain received the lowest perception rate among the participants. Nurses showed a higher positive perception than doctors regarding job satisfaction, stress recognition, and perceptions of management domains. There were statistically significant differences between experiences, workloads, adverse events, and total mean scores of patient safety attitudes. Policymakers and directors can improve the quality of care of patients and patient safety by boosting the decision-making of health care providers on several domains of safety attitudes. Patient safety needs to be improved in hospitals through in-service education, management support, and institutional regulations.

## 1. Introduction

Patient safety is the avoidance of associated adverse events or harms happening from health care [1]. According to the concept of “first, do no harm,” a concept that is involved in essential ethical principles and human rights, a primary emphasis in delivering health care is maintaining patient safety [2]. The National Patient Safety Foundation asserted the importance of patient safety within organizations and kick-started a global impulse toward ensuring preventing injury and appropriate care to all patients [3].

The World Health Organization reported that unsafe patient care is related to a significant increase in adverse medical events throughout the world [3]. In developed countries, estimates suggest that 1 in 10 patients are exposed to injuries during their hospitalization period [1]. In the United States, medical errors are classified as the third cause of death [4]. It was estimated that 6.3 million patient incidences concerning medical care resulted in costs totaling approximately 19,571 million dollars [5]. In Sweden, the National Board of Health reported that more than 9% of patients in somatic care were subject to preventable errors [6]. In developing countries, the potential of adverse events is far above that in developed states. For example, in the Eastern Mediterranean Region, the annual numbers of adverse events are around 4.4 million, and 18% of inpatient admissions were associated with adverse events, with an associated high rate of death and lifelong disability [7]. Another study conducted in Palestinian hospitals found that one in seven patients admitted to hospitals faced medical errors, and 59.3% of these errors were preventable [8]. As these figures demonstrate, health care providers’ patient safety competency must be verified and the rate of adverse events reduced to improve patient outcomes and decrease adverse patient-related events and costs.

Patient Safety 2030 reported that failure to secure patient safety was a significant challenge in providing health services. The report recommended boosting patient care practices of health care providers and their awareness of and attitudes toward patient safety in the coming 15 years [9]. An attitude encompasses human beliefs and behaviors that can impact decisions and shape behavior [10]. Thus, determining the attitudes of doctors and nurses in terms of patient safety can simplify identifying measures directed at enhancing attitudes and promoting better clinical outcomes and organizational competencies [11]. Along these lines, evaluating safety attitudes allows safety aspects to become more apparent and can help enhance a health care setting in which errors are known and handled suitably [4]. Patient education in safety is associated with better patient outcomes and improvements in patient empowerment. Positive patient attitudes about engaging in their own safety integrated with health care provider’s efforts to enhance safety can have synergistic outcomes [12,13].

Several studies have been performed to assess the patient safety attitudes among doctors and nurses and reported that physicians and nurses had negative patient safety attitudes, such as low perceptions of teamwork climate and management domains [6,14]. In a quantitative, descriptive, cross-sectional study conducted in Portugal, the participants reported negative perceptions of the safety climate of patients [15]. In a study conducted in Sweden among health care providers who worked in an emergency department, participants said that strong management commitment was necessary to boost patent safety and create a safety culture [6]. A recent study conducted in Turkey using 290 operating room staff, including doctors and nurses, found that the “safety culture” should be improved, and communication was necessary [16]. A national study in Northern Cyprus found that developing an appropriate safety culture was necessary to ensure patient safety [17]. Another study in Northern Cyprus argued that “healthcare managers and decision-makers should foster patient safety culture through in-service education, management support, institutional regulations, and updated guidelines.” [18]. Thus, establishing an organizational culture emphasizing patient safety and adopting appropriate attitudes and behaviors by healthcare workers is a primary factor for safe care [19]. Although numerous hospitals endeavor to improve patient safety policies, several barriers impede progress and contribute to reducing healthcare performance-related patient safety, such as workload, adverse events, and experience [20,21].

Adverse events are a crucial point of quality in hospitals that will provide information about the lack of quality of care [21]. Adverse events that arise during healthcare delivery are associated with rising morbidity and mortality and are negatively associated with the safety attitudes of health care providers [22]. Previous studies have shown a significant relationship between health care providers and workload [23,24]. Workload significantly impacts healthcare services due to its association with staffing challenges, such as burnout and turnover, mortality, and adverse events [25]. Experience is another factor that may be associated with patient safety attitudes. Several studies showed that experienced healthcare providers have higher patient safety attitudes than less-experienced healthcare providers [14,20,23,25,26].

Further studies are needed regarding patient safety attitudes and associated factors to increase knowledge related to patient safety and increase health care quality. The study’s primary aim was to investigate patient safety attitudes among doctors and nurses. The secondary aim was to explore associations between workload, adverse events, and experience with patient safety attitudes, followed with research questions:What are the patient safety attitudes of doctors and nurses?Is there any difference between the safety attitudes of doctors and nurses?Are there any associations between workload, adverse events, experience, and patient safety attitudes?

## 2. Materials and Methods

### 2.1. Design

The study used a descriptive and cross-sectional design.

### 2.2. Setting and Sample

The study was conducted in two private hospitals in Northern Cyprus. One is a large teaching hospital with 13 inpatient units, 5 critical care units, 8 operating rooms, and more than 200 inpatient beds. The second one has more than 150 beds spread across different units. A total of 450 doctors and nurses working in both hospitals (112 doctors, 338 nurses) were asked to complete patient safety attitude questionnaires; 73 doctors and 246 registered nurses completed the study with a response rate of 75%. The inclusion criteria were included doctors and registered nurses. Diploma holders, students, and trainees did not participate in the study.

### 2.3. Study Tools

Data were collected using a descriptive data tool and The Safety Attitudes Questionnaire (SAQ). Demographic data of the participants such as age, gender, job titles, education level, hospital type, work shifts, and safety education and training were obtained using a descriptive data tool the researchers prepared.

The original version of SAQ was developed by Sexton et al., which has been used in various settings to assess the safety attitudes of healthcare providers across safety competency as well as contribute to recognizing areas that need improvement [27]. In the current study, the Turkish version was used as a data collection tool. The Cronbach alpha ranged from 0.66 to 0.77. [28]. The Turkish version of SAQ used 30 of the 59 original items in the safety attitudes questionnaire. The scale consists of six domains: teamwork climate (6 items), safety climate (7 items), job satisfaction (5 items), stress recognition (4 items), perceptions of management (4 items), and working conditions (4 items). Two items of two-scale dimensions were scored negatively (teamwork item number 2 and safety climate item number 4). Concerning the question “how many errors you have reported in the 12 months,” participants were told that these errors could include any harm to patients, surgical errors, wrong dose, and omission of treatment, missed documentation, falls, and wrong patients. In this study, the instrument’s reliability was assessed with Cronbach’s α, which was 0.74.

### 2.4. Data Collection

The questionnaires were implemented between May and July 2019 with a self-completion method. Before implementing the survey, the data collectors explained the aim of the study, and all participants were provided with an informed consent form, to which they had to assent to participate. Also, they were informed that their participation was voluntary and anonymous. The study results would not impact their annual evaluations, and they could withdraw from the study at any time.

### 2.5. Data Analysis

Data were screened before entry into Statistical Package for the Social Sciences (Version 22.0, SPSS Inc., Chicago, IL, USA). Descriptive statistics (frequency, percentages, and mean) were computed for the demographic data. Statements regarding patient safety attitudes used a 5-point Likert scale with scores ranging from disagree strongly (1) to agree strongly (5). The Likert scale scores were charted as (1 = 0, 2 = 25, 3 = 50, 4 = 75, and 5 = 100) on a 100-point scale. The cut points were computed sum of the total frequency of agree slightly (4) and agree strongly (5), divided by the total respondents’ numbers, and multiplied by 100%. If the total percentage was more than or equal to 75, this indicated a positive response; less than 75 indicates a negative response. Regarding negative responses, reverse coded of negatively items to indicate a positive perception disagree strongly (1), disagree slightly (2). Parametric tests (mean SD, independent sample *t*-tests, and ANOVA) were performed on normally distributed data to examine associations between categorical variables and the relationships between all two groups. The chosen level of significance is *p* < 0.05.

## 3. Results

Most participants from both groups were females (55.3%). The mean age of the doctors was 30.7 ± 8.1, and nurses was 27.3 ± 6.4. Concerning experience, 48.9% of doctors had experiences ranging from 6–10 years, while 49.4% of the nurses had experiences ranging from 1–5 years. Two-thirds of both participants had a workload of 33–48 h/w. Most participants in both groups had not reported errors during the last years. There were no statistically significant between the participants’ in terms of demographic characteristics (Table 1).

Among the six dimensions of the SAQ, the work conditions dimension received the highest positive rate (64.2%), followed by job satisfaction and teamwork climate dimensions (63.9%, 62.4%), respectively. A weak perception of the stress recognition and perceptions of management dimensions of the SAQ was found, with a positive rate of (56.4%, 55.8%) respectively. Safety climate dimension of the SAQ had the lowest positive rate (49.7%) (Table 2).

Regarding patient safety attitudes of the doctors and nurses, results revealed that the total positive frequencies for both participant groups were beneath the positive score (>75%), which indicates the overall perceptions of both participant groups were negative toward patient safety (62.9 ± 22.4; 58.6%). Although not significant statistically, nurses had higher total positive perceptions (63.3%) than doctors (54.3%). There were statistically significant differences between positive responses of the nurses and doctors in terms of job satisfaction, stress recognition, and perceptions of management domains. Nurses showed higher positive perceptions (69.3%) than doctors (58.1%) in terms of the job satisfaction domain (*p* < 0.05). In the stress recognition domain, nurses showed higher positive perceptions (57.4%) than doctors (55.4%) (*p* < 0.05). Nurses also showed higher positive perceptions (68.3%) than doctors (45.4%) in terms of perceptions of the management domain, which was statistically significant (*p* < 0.05) (Table 3).

Table 4 compares the participants’ patient safety attitudes values with the workload, adverse events, and experience. There were statistically significant differences between experiences, workloads and adverse events, and total patient safety attitudes mean scores of the participants (*p* < 0.05). In terms of years of experience, nurses who have 1–5 experience years showed lower patient safety attitudes mean scores (54.7 ± 7.5) than the other age groups (*p* < 0.05). The doctors and nurses who work >48 h/w and 3 more adverse events showed lower patient safety attitudes mean scores than the other groups, which was statistically significant (*p* < 0.05).

Multiple linear regression analysis was carried out to determine the contribution of factors within the patient safety attitude questionnaire. Hospital type above 0.05, were excluded. The overall model was statistically significant and identified two variables (gender, age) explained 16.1% of the variance in the patient safety attitude with F (2, 134) = 9.174 (*p* < 0.001) (Table 5).

## 4. Discussion

This study was conducted among doctors, and nurses in two hospitals in Northern Cyprus, to assess patient safety attitudes perceptions among doctors and nurses and explore associations between workload, adverse events, and experience with patient safety attitudes. This study’s findings will allow medical care institutions to determine aspects that need attention and areas of weakness and strengths associated with patient safety.

The study results indicated that overall patient safety attitudes among doctors and nurses were negative. A possible explanation for negative perception could be that our participants were less experienced and had lower educational qualifications than other studies. Better educated and more experienced staff are more mature and professionally responsible for patient safety [29,30]. They can better understand complications and are more aware of quality of care issues [31,32].

The current study’s findings are consistent with research conducted in Saudi Arabia among doctors and nurses that revealed negative attitudes for all safety domains, which created challenges for developing a safety culture [11]. Another study found that all respondents had negative attitudes towards all safety attitude dimensions [24]. In some international studies, health care professionals revealed a negative safety attitude [6,15]. Negative attitudes may prevent interventions and improvement of care in health care institutions that help maintain patient safety.

Although not significant statistically, nurses exhibited a higher total positive patient safety attitude than doctors, in line with Tunçer and Harmanci [14]. A higher positive patient safety attitude of nurses could be because nurses more closely deal with patients and more often coordinate patient healthcare than physicians do [16]. The lower positive patient safety attitudes of doctors might be because they are less directly involved in daily patient care [33] or patient safety education is absent from many medical school curriculums [34]. Interestingly, doctors showed more positive patient safety attitudes than nurses in several studies [15,33]. Variations in organizational cultures and within different doctors’ specialties might impact patient safety [31].

The work conditions domain had the highest perception rate. Although it was beneath the positive cut-off, it was higher than benchmarking results [28]. This is an interesting finding because many studies have shown that healthcare workers were often dissatisfied with working conditions [27,35]. The better response in the current study to this domain indicated that participants were relatively satisfied with their ergonomics, logistical and technical assistance, and support of new employees. Studies indicated that training and supervising, adequate staffing, and maintaining therapeutic and diagnostic information can create better working conditions [23,36].

Job satisfaction is related to personnel morale, motivations, and work satisfaction in a hospital; hence, it cannot be overlooked. In the present study, job satisfaction had the second-highest positive rating. The results were consistent with the results of several studies [23,30]. Morale and pride in the workplace increase job satisfaction of nurses and doctors. However, shortages and excess workloads inside institutions increase the incidence of medication errors and adverse events [24].

The teamwork climate domain was the third in terms of a positive rate. There were statistically insignificant differences among nurses and doctors. The current finding did not align with other studies [24,36]. Difficulties in speaking up, feeling that asking questions when things are not understood is discouraged, and disagreements not resolved appropriately contribute to a low teamwork rating [26]. A positive work environment is essential because such a work environment, in combination with adequate staffing, enhances care outcomes and creates a safer care setting [37]. Several interventions have been recommended to better a work environment, reduce errors, and improve care. These interventions include improving communication and collaboration between physicians and nurses and being able to ask questions to learn from mistakes to improve the teamwork climate [38].

Stress recognition had a lower positive perception rate but had a slightly higher score in this study than in other studies [23]. Nurses showed a higher positive perception than doctors, which aligned with other studies [26]. The notion that a person can make suitable decisions irrespective of the stress they are subjected to is invalid. Excessive fatigue, tense situations, and workload lead to impaired performance and make people more vulnerable to making errors [32]. Several interventions are required to help this situation. These include regulating the work and rest balance, improving healthy lifestyles through health programs, and creating stress management policies [31].

Perception of the management domain encompasses factors relating to staff management and administrative support. Nurses showed a higher positive perception than doctors in terms of perceptions of the management domain. This result was consistent with several studies [16,23]. Management decisions related to policies may lead to confusion, and staff shortages may induce workload, hostile situations, and ultimately, mistakes [24]. Thus, the policymakers must focus on active communication, shared responsibility, and a suitable employee-patient ratio [38].

The safety climate domain received the lowest perception rate among nurses and doctors. It showed a statistically insignificant difference, reflecting the low impact of safety climate among health care providers in their institutions. Studies conducted in developing countries have shown that health professionals had negative attitudes regarding the safety climate [14,36]. The reason could be that respondents have difficulty discussing errors and lack support. Robello et al. recommended that identifying mistakes, learning from those mistakes, and reducing safety hazards are significant aspects in decreasing patient injury and creating a safety attitude among the staff. [15].

In terms of experience, participants with fewer years of experience showed statistically significant differences and lower patient safety attitudes mean scores, which contradicted other studies [36]. Studies reported that when experience increases, patient safety perceptions also increase [14,26]. Possibly inexperienced and newly graduated healthcare givers suffer from stress regarding practice, which makes them vulnerable to an increased incidence of errors and affects their perceptions of patient safety [39]. Vast numbers of staff with insufficient experience and inadequate concerns for patient safety are a risk to patient care and other health professionals [40].

In this study, the doctors and nurses who worked longer hours and reported more events showed lower positive patient safety attitudes, similar to studies [35,36]. The literature reveals that a higher workload is closely associated with increased adverse events [20]. A high workload adversely affects healthcare professionals, diminishes services competency, and adversely influences patient care [41,42]. If these workload challenges continue for an extended period, they might result in serious harm in the hospital and reduce possible solutions in terms of staff health and patients’ care.

Strengths and limitations of this study:The study’s main strengths are the large sample size and the high response rate from doctors and nurses of different departments and clinics.The study was carried out in only two hospitals, making the generalization of the results problematic.There is a need for further research with different designs like qualitative methodologies to examine patient safety attitudes about health care workers to better understand and improve the quality of health care services.

## 5. Conclusions

This study provides information about the level of perception of patient safety attitudes among nurses and doctors in Northern Cyprus. The results demonstrate a negative perception of all patient safety domains, contributing to identifying areas requiring enhancement and factors impeding the development of a safety culture. The work conditions domain received the highest positive perception rate, and the safety climate domain received the lowest perception rate among participants. In terms of experiences, participants with fewer years of experience showed lower patient safety attitudes mean scores than the more experienced participants. The doctors and nurses who worked longer and reported more events showed lower positive patient safety attitudes.

## Figures and Tables

**Table 1 healthcare-10-00631-t001:** Demographic Characteristics of the Participants (N = 319).

Demographic Characteristics	Doctors		Nurses		Total		*p* Value *
	N	%	N	%	N	%	
Hospital							
First hospital	50	81.5	177	83.4	227	82.1	0.5
Second hospital	23	18.5	69	16.6	92	17.9	
Gender							
Male	33	44.7	103	46.8	136	45.7	
Female	40	55.3	143	53.2	138	54.3	0.9
Age							
20–25 years	22	34.4	127	50.9	149	42.6	
26–30 years	33	46.2	77	35.7	110	40.9	0.8
>31 years	18	19.4	42	13.4	60	16.5	
Mean	30.7 ± 8.1		27.3 ± 6.4				
Years of experience							
1–5 years	13	18.4	120	49.4	133	33.9	
6–10 years	35	48.9	72	33.8	107	41.4	0.27
>11 years	25	32.7	54	16.8	79	24.7	
Workload							
16–32 h/w	12	7.2	10	6.6	22	6.9	
33–48 h/w	50	85.8	195	83.1	200	84.5	0.18
>48 h/w	11	7.0	41	10.3	52	8.6	
Event report within the last year							
No events	66	96.9	210	86.4	276	91.7	
1–2 events	5	2.3	23	8.7	28	5.5	0.12
3 more events	2	0.8	13	4.9	15	2.8	
Total	73	100	246	100	319		

* *p* < 0.05, frequencies test; chi-squared test.

**Table 2 healthcare-10-00631-t002:** Positive Response Percentages of the Participant.

	Positive Responses (>75%) *
Job Satisfaction	63.9
I receive appropriate feedback about my performance	54.7
Hospital management does their job well	66.8
This hospital is a good place to work	60.1
Working in this hospital is like being part of a large family	68.6
This hospital deals constructively with problem personnel	67.5
In this hospital, the moral of the nurses is valued	61.6
I am proud to work at this hospital	56.3
The medical equipment in this office is adequate	62.6
The levels of staffing in this hospital are sufficient to handle the number of patients	63.1
Decision making in this hospital utilizes input from relevant personnel	65.1
I am provided with adequate, timely information about events in the hospital that might affect my work	58.7
Teamwork Climate	62.4
While the patient is undergoing care, other employees help me teamwork	64.1
It is easy for personnel in this hospital to ask questions when there is something that they do not understand	62.5
In this hospital, ethical values are high	59.8
During emergencies, I can predict what other personnel are going to do next	65.7
Disagreements in this hospital are resolved appropriately	57.8
Employees who are really professional do not reflect their personal problems	69.2
In this hospital, teamwork and cooperation are supported among employees	66.1
I am encouraged by my colleagues to report any patient safety concerns I may have	61.5
The culture in this hospital makes it easy to learn from the errors of other	58.9
I saw other staff making mistakes that could harm the patient	62.8
I know the proper channels to direct questions regarding patient safety in this hospital	64.6
The physicians and nurses here work together as a well-coordinated team	66.7
Stress Recognition	56.4
Fatigue impairs my performance during emergency situations **	61.3
When my workload becomes excessive, my performance is impaired **	52.0
Stress caused by personal problems negatively affects performance **	55.6
I am less effective at work when fatigued **	49.3
In this hospital, it is difficult to speak up if I perceive a problem with patient care **	57.6
Working Conditions	64.2
In this hospital, the communication disorders that cause the disruption of the service are widespread **	64.4
Employees often do not care about rules and procedures established in this hospital **	58.4
I am disappointed in my work **	59.6
All employees, including doctors in this unit, do their job well	69.3
I feel exhausted in my work **	67.1
In this hospital, nursing education is appropriately supported	62.9
Safety Climate	49.7
In this hospital, information about event reports is used to ensure the patient safety	51.2
In this hospital, safety reporting systems, patient safety development is utilized	45.1
In this hospital, follow clinical guidelines and evidence-based criteria for patient safety	59.8
In this hospital, we know how to report medical errors when necessary	52.4
In this hospital, patient safety is always considered as priority	48.7
Perceptions of Management	55.8
I would feel safe being treated here as a patient	42.3
In this hospital, medical errors are handled appropriately	51.2
This hospital does a good job of training new personnel	45.1
All personnel in this unit take responsibility for patient safety	59.8
Hospital management does not knowingly compromise the safety of patients	52.4
Hospital management supports my efforts to ensure patient safety	48.7
All information about diagnosis and treatment decisions is routinely given to me	51.9

* Positive Responses of Safety Attitudes Questionnaire; ** Negative statements.

**Table 3 healthcare-10-00631-t003:** Attitudes of participants related to patient safety.

Safety Attitudes Domains	Doctors	Positive Response	Nurses	Positive Response	Total Mean (SD)	Total Positive Responses (>75%) *	*p* Values **
	Mean (SD)	(>75%)	Mean (SD)	(>75%)			
Teamwork Climate	66.2 ± 11.9	57.6	68.1 ± 12.1	65.4	67.1 ± 9.1	62.4	0.44
Job satisfaction	69.3 ± 28.4	58.1	74.7 ± 21.1	69.3	71.5 ± 15.4	63.9	0.01
Work Conditions	60.4 ± 30.8	66.4	64.0 ± 27.4	62.0	62.2 ± 25.7	64.2	0.21
Stress Recognition	59.5 ± 28.4	55.4	56.0 ± 29.7	57.4	57.4 ± 21.3	56.4	0.04
Perceptions of Management	49.3 ± 32.0	45.4	53.6 ± 34.2	68.3	50.3 ± 30.6	55.8	0.0
Safety Climate	65.0 ± 27.4	42.9	71.0 ± 26.4	55.8	67.5 ± 22.8	49.7	0.14
Total	61.6 ± 21.4	54.3	64.5 ± 23.6	63.3	62.9 ± 22.4	58.6	0.19

* The comparisons were made between the positive responses. ** Independent sample *t*-tests *p* < 0.05;

**Table 4 healthcare-10-00631-t004:** Comparison of the Participants’ Patient Safety Attitudes perceptions with Workload, Adverse events, and Experience.

Descriptive Characteristics	Doctors		Nurses		Total SAQ Mean Score ± SD	*p*-Value *
	Total SAQ Mean Score ± SD	*p*-Value	Total SAQ Mean Score ± SD	*p*-Value		
Years of experience						
1–5 years	55.4 ± 9.1	0.11	54.7 ± 7.5	0.03	54.0 ± 5.1	0.01
6–10 years	57.1 ± 7.3		59.9 ± 4.4		55.8 ± 4.5	
>11 years	61.9 ± 10.3		60.7 ± 3.8		60.1 ± 5.2	
Workload						
16–32 h/w	45.3 ± 9.2	0.03	53.9 ± 2.1	0.00	49.8 ± 3.8	0.001
33–48 h/w	41.3 ± 5.7		52.1 ± 9.1		46.7 ± 7.2	
>48 h/w	40.1 ± 7.4		50.6 ± 3.1		45.9 ± 4.9	
Adverse events within the last year						
No events	55.4 ± 5.1	0.006	51.3 ± 1.1	0.00	52.9 ± 3.1	0.00
1–2 events	51.6 ± 3.2		49.2 ± 2.7		50.0 ± 2.2	
3 more events	50.2 ± 2.4		49.0 ± 7.5		49.3 ± 5.7	

* ANOVA.

**Table 5 healthcare-10-00631-t005:** Multiple linear regression analysis results: factors associated with patient safety attitude questionnaire score.

	Patient Safety Attitude		
	ΔR2	β	*p*
Hospital type		0.159	0.06 *
Gender		0.124	0.002 *
Age	0.161	0.111	0.001 **

* *p* < 0.05; ** *p* < 0.01 Multiple linear regression.

## Data Availability

The data presented in this study are available on request from the corresponding author.

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
