# Peer review of "Patient Safety Attitudes among Doctors and Nurses: Associations with Workload, Adverse Events, Experience"

_healthcare, 2022, doi:10.3390/healthcare10040631_

Round 1

Reviewer 1 Report

Al-Mugheed et al. investigate patient safety attitudes among doctors and nurses and explore associations between workload, adverse events, and experience with patient safety attitudes. It is quite interesting study. In my opinion language requires some improvement. I have also some methodological remarks-in methodological section authors used for the patients safety attitudes a 5-point Likert scale with scores ranging from disagree strongly (1) to agree strongly (5). The Likert scale scores were charted as (1=0, 2=25, 3=50, 4=75, and 5=100) on a 100-point scale. I think authors should also considered to perform statistics for raw data. I have also minor comment to add also to introduction section short point about impact of patient’s education on the safety attitude (Neurol Neurochir Pol 2021;55(6):582-591.).

Author Response

Point 1: Al-Mugheed et al. investigate patient safety attitudes among doctors and nurses and explore associations between workload, adverse events, and experience with patient safety attitudes. It is quite interesting study. In my opinion language requires some improvement. I have also some methodological remarks-in methodological section authors used for the patients safety attitudes a 5-point Likert scale with scores ranging from disagree strongly (1) to agree strongly (5). The Likert scale scores were charted as (1=0, 2=25, 3=50, 4=75, and 5=100) on a 100-point scale. I think authors should also considered to perform statistics for raw data. I have also minor comment to add also to introduction section short point about impact of patient’s education on the safety attitude (Neurol Neurochir Pol 2021;55(6):582-591.).

Response 1

  • language requires some improvement. The language of manuscript were improvement.
  • I think authors should also considered to perform statistics for raw data. The raw data were added in table 2.
  • patient’s education on the safety attitude. The section short point were added in the introduction pag 2 line 70-73.

Reviewer 2 Report

Thank you for the opportunity to review the article entitled: Patient Safety Attitudes among Doctors and Nurses: Associations with Workload, Adverse Events, Experience

The work is interesting and of good quality.

I have some comments.

Table 1 describes the demographic characteristics of the participants. It could be interesting to know if there are differences among the participants in the variables shown in the table. In addition, the authors should add a table footer where the tests used are accurately described.

In table 2, it is necessary to add a table footer indicating which statistical tests are used.

It could be interesting to carry out an adjusted analysis in which it is possible to jointly compare how the responses of health personnel are affected on the levels of perception of patient safety.

Author Response

Point 1: Thank you for the opportunity to review the article entitled: Patient Safety Attitudes among Doctors and Nurses: Associations with Workload, Adverse Events, Experience

The work is interesting and of good quality.

I have some comments.

Table 1 describes the demographic characteristics of the participants. It could be interesting to know if there are differences among the participants in the variables shown in the table. In addition, the authors should add a table footer where the tests used are accurately described.

In table 2, it is necessary to add a table footer indicating which statistical tests are used.

It could be interesting to carry out an adjusted analysis in which it is possible to jointly compare how the responses of health personnel are affected on the levels of perception of patient safety.

Response 1

  • Table 1 describes the demographic characteristics of the participants. It could be interesting to know if there are differences among the participants in the variables shown in the table. The differences among the participants were performed in Table 1
  • In addition, the authors should add a table footer where the tests used are accurately described. The test named were added.
  • In table 2, it is necessary to add a table footer indicating which statistical tests are used. The test naemed were added.
  • It could be interesting to carry out an adjusted analysis in which it is possible to jointly compare how the responses of health personnel are affected on the levels of perception of patient safety. Th Multiple linear regression were used in Table 5.

Round 2

Reviewer 1 Report

Authors respond to all my comments. I have no further remarks. I suggest acceptance of the paper